# Ocean forced evolution of the Amundsen Sea catchment, West Antarctica, by 2100

Alanna V. Alevropoulos-Borrill[1,a], Isabel J. Nias[1,b], Antony J. Payne[1], Nicholas R. Golledge[2], Rory J. Bingham[1]

[1]Centre for Polar Observation and Modelling, School of Geographical Sciences, University of Bristol, University Road, Bristol, BS8 1SS, UK
[a]Now at Antarctic Research Centre, Victoria University of Wellington, Wellington, 6012, New Zealand
[b]Now at NASA Goddard Space Flight Center, Greenbelt, MD, USA
[2]Antarctic Research Centre, Victoria University of Wellington, Wellington, 6012, New Zealand

*Correspondence to*: Alanna Alevropoulos-Borrill (alanna.alevropoulosborrill@vuw.ac.nz)

**Abstract.** The response of ice streams in the Amundsen Sea Embayment (ASE) to future climate forcing is highly uncertain. Here we present projections of 21st century response of ASE ice streams to modelled local ocean temperature change using a subset of Coupled Model Intercomparison Project (CMIP5) simulations. We use the BISICLES adaptive mesh refinement (AMR) ice sheet model, with high resolution grounding line resolving capabilities, to explore grounding line migration in response to projected sub-ice shelf basal melting. We find a contribution to sea level rise of between 2.0 cm and 4.5 cm by 2100 under RCP8.5 conditions from the CMIP5 subset, where the mass loss response is linearly related to the mean ocean temperature anomaly. To account for uncertainty associated with model initialisation, we perform three further sets of CMIP5 forced experiments using different parameterisations that explore perturbations to the prescription of initial basal melt, the basal traction coefficient, and the ice stiffening factor. We find that the response of the ASE to ocean temperature forcing is highly dependent on the parameter fields obtained in the initialisation procedure, where the sensitivity of the ASE ice streams to the sub-ice shelf melt forcing is dependent on the choice of parameter set. Accounting for ice sheet model parameter uncertainty results in a projected range in sea level equivalent contribution from the ASE of between -0.02 cm and 12.1 cm by the end of the 21st century.

## 1. Introduction

The contribution of the Antarctic Ice Sheet is the greatest uncertainty in estimates of projected global mean sea level rise (Church et al., 2013; Schlegel et al., 2018). The Amundsen Sea Embayment (ASE) sector, West Antarctica, has been identified as a focal region for mass loss (McMillan et al., 2014; Shepherd et al., 2012, 2018), draining one third of the West Antarctic Ice Sheet (Mouginot et al., 2014). Both observational (Rignot et al., 2014; Smith et al., 2017) and modelling studies (Favier et al., 2014; Gladstone et al., 2012; Golledge et al., 2019; Ritz et al., 2015) have inferred that the region is susceptible to rapid and widespread retreat through marine ice sheet instability (MISI) given that the ASE ice streams are

grounded on retrograde bedrock below sea level (Schoof, 2010; Weertman, 1974). Ocean forced sub ice-shelf basal
melting acts to reduce the buttressing effect of ice shelves in the ASE, altering the longitudinal stress balance and causing a
speed up of flow (Gudmundsson, 2013). Once initiated, flow acceleration leads to increased thinning and subsequent
grounding line retreat, driving further mass loss through increased flux, where flux increases as a function of thickness at the
grounding line (Schoof, 2007). The stability of the ASE ice streams is therefore largely dependent on ocean forcing and
subsequent sub-shelf melting (Jacobs et al., 2012; Jenkins et al., 2018; Pritchard et al., 2012).


Ocean forcing in the ASE differs from much of the Antarctic Ice Sheet due to a combination of the continental topography,
the depth of the thermocline and the Pacific Ocean climatology, namely the proximity of the Antarctic Circumpolar Current
to the continental shelf (Pritchard et al., 2012; Turner et al., 2017). In the ASE, atmospheric and oceanic mechanisms drive
an upwelling of warm Circumpolar Deep Water (CDW), reaching up to 4°C above the *in situ* melting point, which is routed
toward the grounding lines of the ASE glaciers through dendritic bathymetric troughs (Nakayama et al., 2014; Thoma et al.,
2008; Turner et al., 2017; Webber et al., 2017). It is widely accepted that CDW is responsible for observed high rates of
melting beneath ASE ice shelves (eg. Pritchard et al., 2012; Walker et al., 2013) where periods of CDW intrusion in the ASE
coincide with a speed up of glacier velocity (Parizek et al., 2013; Payne, 2007; Shepherd et al., 2012), making the presence
of this water mass on-shelf an important control on ice dynamics and regional mass loss. Observations have shown an
increase in the quantity of CDW on-shelf in the ASE (Schmidtko et al., 2014), and projections show that this will continue in
the future, with the increased positive phase of the Southern Annular Mode and subsequent strengthening of circumpolar
westerlies acting to drive CDW on-shelf (Bracegirdle et al., 2013; Spence et al., 2014).

In this investigation, we first identify a subset of Coupled Model Intercomparison Project Phase 5 (CMIP5) atmosphere-
ocean general circulation models (AOGCMs) that best reproduce historical observations of Southern Ocean temperature.
Using this subset, we then use the RCP8.5 projections of ocean temperature anomalies in the ASE from 2017-2100 to
parameterise a melt rate forcing for the BISICLES ice sheet model. The use of separate projections from individual
AOGCMs provides indication as to the range of uncertainty associated with the choice of modelled ocean temperature
projection and thus uncertainty associated with the applied ocean forcing. Finally, we explore the uncertainty associated with
the model initialisation procedure through additional experiments with perturbed sets of the spatially varying parameter
fields obtained in the initialisation procedure. The findings provide fresh insight into the projected migration of the
grounding lines of the ASE ice streams when represented by a model with adapting fine grid resolution adjacent to the
grounding line. Additionally, we present new, constrained, estimates of the projected sea level contribution from the ASE in
response to CMIP5 projected regional ocean forcing under the RCP8.5 'business-as-usual' scenario.


## 2. CMIP5 subset

The CMIP5 ensemble consists of 50 AOGCMs and earth system models (ESMs) from 21 modelling groups (Taylor et al., 2012), providing a valuable resource for exploring the projected future evolution of the climate under varying future emission scenarios. Biases in the representation of climatological features in the Southern Ocean have been widely
investigated (Bracegirdle et al., 2013; Hosking et al., 2013; Little and Urban, 2016; Meijers et al., 2012; Sallée et al., 2013a; Sallée et al., 2013b), and individual model representation of observed climate varies largely across the ensemble (Flato et al., 2013). Comparing the output of AOGCMs against climatological observations provides a means by which we can investigate biases, assess model performance (Gleckler et al., 2008) and identify models that best reproduce observed climate in the Southern Ocean. Assuming performance is temporally consistent, projections of climate produced by well-
performing models can be utilised in experiments establishing future basal melt rates (Naughten et al., 2018); thus providing an input forcing for standalone ice sheet models.

### 2.1 CMIP5 model assessment

To identify the CMIP5 models which best reproduce Southern Ocean climate, we use the root mean square error (RMSE)
performance metric, which is common practice in model evaluation (Gleckler et al., 2008; Little and Urban, 2016; Naughten et al., 2018). We compare modelled monthly CMIP5 output of ocean potential temperature below 30°S from January 1979 to December 2016 against the Hadley Centre EN4.2.1 dataset of monthly ocean potential temperature (Good et al., 2013; downloaded 08/02/2018) over the equivalent period. The observational data is corrected for biases following Gouretski and Reseghetti (2010) methods, and quality control flags are used to nullify potentially unreliable observations from the dataset.
Models are evaluated over the whole Southern Ocean on the basis that teleconnections across the Pacific Ocean have been shown to directly influence ocean heat transport in the ASE (Steig et al., 2012). Furthermore, there are limited observations in the ASE (Mallett et al., 2018), limiting the validity of regional evaluation.

Given that the historical period defined by the CMIP5 ensemble ends in December 2005, we use ocean potential temperature
projections forced with both RCP2.6 and RCP8.5 to make up the remaining decade, from January 2006 to December 2016, of the observational period. This restricts analysis to the 27 AOGCMs with projections for both RCP2.6 and RCP8.5 scenarios. Given the differences in model resolution and depth levels, we perform bilinear interpolation of the gridded model output onto the location of the observational dataset and further depth-wise linear interpolation, giving the modelled equivalent of each in situ temperature profile. We calculate two separate RMSE scores for each model, using both RCP2.6
and RCP8.5 which we average to give an overall RMSE for each CMIP5 AOGCM.

## 2.2. Subset selection

Based on the mean RMSE for both RCP2.6 and RCP8.5 simulations of ocean temperature in the Southern Ocean, we select the six AOGCMs with the lowest score, and thus the most realistic representations of observed ocean potential temperature in the Southern Ocean. Models bcc-csm1-1, CanESM2, CCSM4, CESM1-CAM5, MRI-CGCM3, NorESM1-ME comprise our subset. Additionally, we include the two additional models in the subset which have the highest (GISS-E2-R) and lowest (GISS-E2-H) mean projected temperature anomalies over the 21[st] century, local to the ASE (see below for zonal calculation), in order to capture the full range of projected temperatures on-shelf in the ASE across the CMIP5 ensemble.

## 2.3. Ocean temperature in the ASE

We explore modelled and observed ocean temperature in the ASE by averaging ocean temperature over the 400-700 m layer and then averaging from 103-113°W and 72-74°S to cover the ASE continental shelf. Depths of 400-700 m are chosen to represent the depth of CDW on-shelf (Arneborg et al., 2012; Little and Urban, 2016; Nakayama et al., 2014; Thoma et al., 2008; Webber et al., 2017). Of the models that best reproduce temperature over the Southern Ocean, the range in modelled temperature on-shelf in the ASE is ~2°C (fig. 1). Whilst no model is able to capture the range of observed variability in ocean temperature on-shelf, which has been shown to oscillate by up to 2°C (Jenkins et al., 2018), the collective model output captures the overall range in observed ocean temperature. Of the CMIP5 models in the subset, bcc-csm1-1, CanESM2, CCSM4 and NorESM1-ME most closely reproduce observations on-shelf in the ASE. Analysis is, however, limited by the number of observations in the region due to seasonal dependence of ship access and lack of mooring-based observations (Kimura et al., 2017) meaning seasonal variability is not fully captured by observations in this, or other, data sets (Mallett et al., 2018). As no single model captures the observed ocean temperature variability on-shelf, we argue that the use of a subset as opposed to an individual model forcing is advantageous as it covers a greater range of possible ocean temperatures on-shelf.

## 2.4. CMIP5 Ocean Temperature Projections

Having identified a subset of AOGCMs, we explore the 21[st] century ocean temperature projections in the ASE as modelled by each subset member. To gain uniformity of AOGCM resolution, the projection data from each CMIP5 subset member is bilinearly interpolated onto a uniform 1°x1° horizontal grid. To prescribe a mean ocean temperature forcing for our ice sheet model experiments, we calculate the mean annual ocean potential temperature anomalies in the ASE (fig. 2). Anomalies are calculated relative to the 2006-2016 temporally averaged mean for the ASE over the 400-700 m depth-averaged layer. The ASE is again defined as the region between 103-113°W and 72-74°S, a southern limit is established in order to remove regions where an ice shelf would reside as no ice shelf cavity is represented in the CMIP5 ensemble (Naughten et al., 2018).

Whilst projected ocean temperatures under the RCP2.6 scenario have been obtained, the projected anomalies lie within the range of ocean temperature projections for the RCP8.5 scenario. As this investigation is interested in exploring a range of temperatures, RCP8.5 projections alone have been used in the remainder of the study.

The modelled range of ocean temperature anomalies under the RCP8.5 scenario diverge over the 21$^{st}$ century with a 2.2°C range in anomalies by 2100. With the exception of MRI-CGCM3, all models project a temperature increase over the 21$^{st}$ century, relative to the 2006-2016 mean, in response to the business-as-usual scenario. Ocean warming captured by the subset is broadly consistent with the 0.66°C full CMIP5 ensemble mean warming over the 21$^{st}$ century in the ASE (Little and Urban, 2016). The models projecting the largest increase in temperature over the 21$^{st}$ century, namely GISS-E2-R, CanESM2 and bcc-csm1-1, underestimate observed temperature in the ASE during the observational period (fig. 1). Further, the models with warm biases over the observational period, MRI-CGCM3, CESM1-CAM5 and GISS-E2-H, project the lowest temperature change over the projection period.

We attribute the projected temperature changes to modelled changes in the quantity of CDW on-shelf in the ASE (fig. 3). The behaviour of the models can be characterised by the pattern of temperature change in the Pacific sector of the Southern Ocean, where models display either a localised warming of over 1°C in the ASE or a regional warming of a lower magnitude, below 0.5°C. Models exhibiting local increases of temperature in the ASE over the projection period have broadly captured on-shelf temperature over the observational period (fig. 1); these are most notably bcc-csm1-1, CanESM2, CCSM4, and NorESM1-ME. We infer the projected localised warming over the 21$^{st}$ century to be a result of increased incursion of the CDW layer on-shelf in the ASE. Increased CDW presence in the ASE has been observed over the last three decades (Schmidtko et al., 2014), a trend which is expected to continue in the 21$^{st}$ century as a result of a strengthening of the circumpolar westerlies that are responsible for delivering warm CDW towards the ASE continental shelf (Bracegirdle et al., 2013; Gille, 2002; Meijers et al., 2012; Sallée et al., 2013b; Spence et al., 2014).

In contrast, the models which overestimate temperatures over the observational period, namely CESM1-CAM5, GISS-E2-H and MRI-CGCM3, do not display localised future warming in the ASE, instead showing a muted regional warming. We hypothesise two possible explanations for this overestimation of observed temperature: either through modelled presence of a warm CDW layer on-shelf that does not change in depth over the course of the projection period resulting in little to no change in mean ocean temperature; or a lack of representation of the CDW incursion mechanism that therefore precludes additional modelled upwelling or incursion.

## 3. BISICLES configuration and CMIP5 forced experiments

### 3.1. Model description and equations

To explore the evolution of the ASE in response to CMIP5 forced sub-ice shelf melt, we use the BISICLES ice flow model. BISICLES is based on the vertically integrated flow model by Schoof and Hindmarsh (2010) which includes longitudinal and lateral stresses, in addition to a simplification of vertical shear stress which is better applied to ice shelves and streams (Cornford et al., 2013; Schoof, 2010). It uses adaptive mesh refinement (AMR) to provide fine resolution near the grounding

line and a coarser resolution elsewhere. For the simulations performed in this study, we use five resolution levels with mesh grid spacing of $\Delta x^l = 2^{-l} \times 4000m$, where $l$ is an integer between 0 and 4, giving a finest resolution of 250m at the grounding line.

Applying mass conservation to ice thickness and horizontal velocity $u$ gives

$$\frac{\partial h}{\partial t} + \nabla \cdot (uh) = M_s - M_b,$$

(1)

where $M_s$ denotes surface mass balance and $M_b$ is the basal melt rate, which, when discretised, is applied solely to cells in which ice is floating.

Upper surface elevation $s$ is dependent on ice thickness $h$ and bedrock elevation $b$, given that ice is assumed to be in hydrostatic equilibrium

$$s = \max\left[h + b, \left(1 - \frac{\rho_i}{\rho_w}\right)h\right],$$

(2)

where $\rho_i$ and $\rho_w$ describe the respective densities of ice and water.

A two-dimensional stress balance equation is also applied, where the vertically integrated effective viscosity $\dot{\varphi}\bar{\mu}$ is obtained from both the stiffening factor $\varphi$ and a vertically varying effective viscosity $\mu$, which was derived from Glen's flow law. The

stress balance equation is therefore formulated as

$$\nabla \cdot [\varphi h \bar{\mu}(2\dot{\boldsymbol{\varepsilon}} + 2tr(\dot{\boldsymbol{\varepsilon}})\boldsymbol{I})] + \tau_b = \rho_i gh\nabla s.$$

(3)

in which the horizontal strain rate tensor is described by

$$\dot{\varepsilon} = \frac{1}{2}[\nabla u + (\nabla u)^T]. \tag{4}$$

The vertically varying effective viscosity $\mu$ includes representation of vertical shear strains and, given that the flow rate exponent $n = 3$, satisfies

$$2\mu A(4\mu^2\dot{\varepsilon}^2 + |\rho_i g(s-z)\nabla s|^2) = 1 \tag{5}$$

where the temperature rate dependent factor $A(T)$ is calculated using the formula described by Cuffey and Paterson (2010). Uncertainty in both temperature $T$ and $A(T)$ is addressed by $\varphi$. The basal traction coefficient $C$ is assumed to satisfy a non-linear power law, where m = 1/3

$$\tau_b = \begin{cases} -C|u|^{m-1}u & h\dfrac{\rho_i}{\rho_w} > b \\ 0, & otherwise \end{cases}. \tag{6}$$

The initial and applied basal melt rate is parameterised so that it is spatially varying with melt concentrated closest to the grounding line according to the following equation

$$M_b(x,y,t) = \begin{cases} M_G(x,y)p(x,y,t) + M_A(x,y)(1 - p(x,y,t)), & floating \\ 0, & grounded \end{cases} \tag{7}$$

where $p(x,y,t)=1$ at the grounding line which then decays exponentially with increasing distance from the grounding line,

$$p - \lambda^2\nabla^2 p = \begin{cases} 1, & grounded \\ 0, & elsewhere \end{cases}, \tag{8}$$

with $\nabla p \cdot n = 0$ as a boundary condition.

## 3.2. Input data

To solve the equations described above, the BISICLES ice sheet model requires numerous input data, which we find from a number of existing studies. Surface elevation ($s$) and surface mass balance ($M_s$) are obtained from Bedmap2 (Fretwell et al., 2013) and we use a 3D temperature field from a higher order model (Pattyn, 2010). The remaining variables ($C$, $\varphi$, $h$, $b$, and $M_b$) are obtained from the results of an initialisation procedure of BISICLES performed by Nias et al. (2016). Of these parameters, the basal traction coefficient ($C$) and viscosity stiffening factor ($\varphi$) are found by solving an optimisation problem

which minimises the mismatch between modelled ice-surface speed and the observed speed from Rignot et al. (2011). Here we use the ice thickness ($h$) and a modified bed topography ($b$) developed by Nias et al. (2016) which was found by modifying BedMap2 using an iterative procedure to smooth inconsistencies in the modelled flux divergence. The initial sub-shelf melt rate ($M_b$) is also calculated through this iterative procedure (Nias et al., 2016) to ensure the melt rate at the beginning of the simulation is consistent with present day observations and matches observed thinning at the grounding line.

Further, the model has been run with a calving front fixed at its initial position.

## 3.3. CMIP5 melt rate forcing

We convert the CMIP5 projections of ocean temperature into a mean additional ocean sub-shelf melt forcing using the linear relationship between temperature anomaly and ice shelf melting which is approximated for the ASE (Rignot and Jacobs,

2002), where an additional 0.1°C temperature increase results in an increase of 1 m/a to the basal melt rate. The CMIP5 AOGCM forcing data that we use are relatively coarse in their spatial resolution and also do not capture sub-ice shelf oceanographic conditions. Consequently, we are unable to accurately incorporate the spatial complexity of ocean temperature variability that exists in the ASE (c.f. Turner et al., 2017). Given that our input data better reflect regional rather than local-scale oceanic changes, we force our simulations with spatially-averaged CMIP5 temperature anomalies.


The additional sub-shelf melt forcing is applied to the model using a distance decay function with the greatest melt rates located at the grounding line to capture some of the spatial distribution of melt (Payne, 2007). We use a grounding line proximity parameter $p$ as a multiplier, where $p = 1$ at the grounding line and decays exponentially with increasing distance. In the 1D case, $p(x) = \exp(-x/\lambda)$ where $\lambda$ is a scale of 10,000 m. The mean additional forcing is applied onto a 2D

spatially varying field, smoothed to match the pattern of melt obtained during the model initialisation procedure.

The simplified distance dependent melt parameterisation employed in this investigation was chosen in order to maintain continuity with the Nias et al., (2016; 2019) studies. Our parameterisation neglects the effect of overturning circulation within an ice shelf cavity in addition to the ice shelf cavity geometry and presence of meltwater plumes which influence the

pattern of sub-ice shelf basal melting (Dinniman et al., 2016). Whilst more complex parameterisations attempt to incorporate

these mechanisms (e.g. Lazeroms et al., 2018; Reese et al., 2018), no parameterisation is yet able to replicate known patterns of sub-ice shelf melting (Favier et al., 2019). Furthermore, the uncertainty associated with the magnitude of the future forcing exceeds that associated with the parameterisation of sub-shelf melting (Holland et al., 2019), justifying the use of the simplified parameterisation employed in this investigation.


## 3.4. Parameter selection

We investigate the impact of parameter uncertainty on the response of the ASE to the CMIP5 ocean forcing by selecting members of a perturbed parameter ensemble performed by Nias et al. (2016), which hereafter we will refer to as the N16 ensemble. Here we will briefly describe the N16 ensemble, before explaining our selection process. As described above, the

initialisation procedure performed by Nias et al. (2016) produces three optimal, spatially-varying fields of the unknown parameters of basal traction coefficient $C$, ice stiffening factor $\varphi$, and initial basal melting $M_b$ over the ASE catchment. The N16 ensemble explores the influence of uncertainty in these parameters on the modelled mass evolution and grounding line migration in the ASE by scaling the optimal parameter fields between a halving and a doubling and proceeding to sample these scaled fields using a Latin Hypercube. The resulting unique combinations of scaled parameters are referred to in this

investigation as parameter sets. For each perturbed parameter set, a 50-year BISICLES simulation was performed and the change in volume above floatation (VAF) was used to calculate a sea level equivalent (SLE) contribution. This was done for each combination of two geometries (modified and unmodified Bedmap2) and two sliding laws, giving a total of 284 simulations.

In order to explore the role of parameter uncertainty in our study, we select three sets of perturbed parameter fields from the N16 ensemble, in addition to the optimum. To represent a crude 90% confidence from the variation of parameters, we select the parameter combinations that generated a high-end, median and low-end SLE contribution over a 50-year transient experiment in the absence of additional forcing. We identify the parameter sets that most closely produce the 5[th] and 95[th] percentile of a calibrated probability density function of the N16 ensemble, as described in Nias (2017). In this investigation

we solely consider the simulations with the non-linear sliding law and modified bedrock. For each parameter set, we perform simulations forced with the CMIP5 ocean temperature projections parameterised as a sub-ice shelf melt rate. We present the scaling factors for the four parameter sets used in this investigation (table 1). The scaling factors describe the level of perturbation for each of the spatially varying parameter fields within each of the four parameter sets where a halving is 0, the optimum is 0.5 and doubling is 1. When discussing the outcome of the results we will use these values as a relative

comparison.

### 3.5. Experimental design

We perform regional simulations of the ASE sector on the domain defined in Cornford et al., (2015). For each of the four different parameter sets, we use parameterised sub ice-shelf melt rates for each of the eight CMIP5 subset members. An additional control experiment is performed for each of the four parameter sets. The control experiment has no additional melt forcing and therefore the results capture the dynamical ice response to present conditions. A total of 36 experiments are performed. The following results section firstly describes the results from the optimum parameter set, followed by the results of the experiments using the three perturbed parameter sets.

For our simulations of future mass evolution of the ASE in response to changing ocean temperature forcing, we choose to keep the atmospheric forcing constant due to the small effect of surface mass balance changes on ice stream dynamics (Seroussi et al., 2014), particularly on the timescales we explore in this investigation. Furthermore, ocean forced sub-shelf melting elicits an immediate response to the upstream ice dynamics (Seroussi et al., 2014) making this the focus of our work.

## 4. Results

### 4.1. Optimum parameter set

Our projections show that by the end of the 21$^{st}$ century the CMIP5 forced sub-ice shelf melting in the ASE will lead to a contribution to global mean sea level of 2.0 – 4.5 cm under the RCP8.5 scenario. The range in SLE in response to each CMIP5 sub-ice shelf melt rate reflects the magnitude of the applied forcing (fig. 4), where the experiments forced with CMIP5 models that project the most extreme temperature change result in the greatest overall mass contribution over the 21$^{st}$ century. The variation in response according to AOGCM forcing indicates a strong dependence of ASE mass loss on sub-shelf melting, consistent with existing literature (Gudmundsson et al., 2019; Pritchard et al., 2012). The most extreme response is a result of the GISS-E2-R projected ocean melting in the ASE which results in 4.5 cm of sea level rise. The model that projects the lowest magnitude ocean temperature forcing, MRI-CGCM3, projects a contribution of 2.0 cm by 2100 despite having a negligible temperature change at the end of the 21$^{st}$ century relative to present day. In contrast, the contribution from the control experiment indicates a committed 2.2 cm contribution to sea level rise in response to recent past and present day forcing. The SLE contribution over the projection period is nonlinear for models with more extreme forcing, which reflects the projected nonlinear increase in ocean potential temperature (fig. 2).

Each of the nine experiments project grounding positions in 2100 which are retreated relative to the initial grounding line positions (fig. 5). The response of the individual ice streams to the varying ocean melt forcings differs as a result of their varying topographic confinements and differing ice dynamics (Nias et al., 2016). Despite the differing magnitudes of the CMIP5 model forcings, the PIG grounding line migrates 25 km upstream from its initial position for all experiments except

MRI-CGCM3 and the control experiment where retreat is 11 km, likely controlled by the steep deepening of the bed over the initial 10 km upstream of the initial grounding line (Vaughan et al., 2006). Stabilisation of the grounding line 25 km upstream of its initial location is indicative of local topographic maxima at this position (Vaughan et al., 2006) and substantial prograde slope evident in the modified Bedmap2 topography described in the N16 study. We infer from the results that, using the optimum parameter set, grounding line migration over the 21[st] century is relatively insensitive to the magnitude of additional forcing, as illustrated by the equivalent grounding line positions. The results from the control experiment denote the projected grounding line migration should climate conditions remain constant, and therefore reveal the committed sea level contribution from the ASE in response to current climate.

Across the model subset, the Thwaites Glacier grounding line is projected to both retreat and lengthen over the 21[st] century, with a greater retreat occurring in the eastern side of the main trunk. A lengthening of the grounding line occurs due to the widening of the ice stream trunk upstream of the grounding line. In response to the varying forcings, the Thwaites Glacier grounding line experiences approximately the same extent of grounding line migration which is clustered at points across the main trunk, showing a level of insensitivity to applied forcing. The exception to this grounding line position is illustrated by the GISS-E2-R forced experiment where migration of the Thwaites Glacier grounding line is marginally greater than for the remaining models. The relative insensitivity of Thwaites Glacier is consistent with previous modelling studies (Tinto and Bell, 2011) which may suggest that the buttressing effect of the unconfined ice shelf is minimal and varying magnitudes of sub-shelf melting have a lesser control on the grounding line position (Parizek et al., 2013). Furthermore, retreat to the same position upstream would indicate that this is a position of stability, where the grounding line is pinned, likely reflecting the presence of a topographic rise. The fact that migration and lengthening of the grounding line occurs even in the control experiment demonstrates that grounding line retreat over the 21[st] century is almost certain.

Grounding line retreat of the Pope, Smith, and Kohler (PSK) ice streams is dependent on the magnitude of the CMIP5 sub-ice shelf melt forcing applied. The most extreme forcing, the GISS-E2-R forced experiment, results in almost complete loss of grounded area of the small ice streams by the end of the 21[st] century, whilst the control experiment results in grounding line retreat of only ~20 km. The variation in grounding line positions in 2100 indicates that the PSK ice streams are sensitive to the magnitude of ocean forcing due to the buttressing provided by the narrow embayment of the ice streams and the confined Crosson and Dotson ice shelves (Konrad et al., 2017). As the ice streams are relatively small compared with their neighbours, almost complete loss of the present ice streams could occur over the 21[st] century, even in the absence of additional ocean forcing (Scheuchl et al., 2016).

## 4.2. Perturbed parameter sets

The range in volume above floatation change from the subset of experiments results in a -0.02 - 1.4 cm SLE contribution for the low-end parameter set, 2.6 - 8.6 cm for the median parameter set and 5.4 - 12.1 cm for the high-end parameter set. As illustrated by the differing range of SLE contributions across the four parameter sets, the sensitivity of the ASE to different additional sub-shelf melt forcings varies with differing spatially varying parameter fields. Again, the magnitude of mass loss is proportional to the magnitude of the applied forcing for each of the CMIP5 forced experiments, and this relationship is

consistent across the three perturbed parameter sets.

Experiments configured with the low-end parameter set result in the most modest grounding line retreat across the ASE ice streams (fig 5). The PIG grounding line is projected to retreat ~14 km upstream of the main trunk for each of the CMIP5 forced experiments, with retreat into the southwestern tributary occurring in some scenarios in response to the different

forcing magnitudes. The projected grounding line position of Thwaites glacier by the end of the 21st century for the low-end parameter set is most equivalent to the present-day position, experiencing minimal retreat with only minor variation between the different CMIP5 forced experiments. Of the ASE ice streams, the Thwaites Glacier grounding line position varies most in comparison to the optimum. Similar to the optimum parameter set experiments, the PSK grounding line retreat differs considerably in response to the varying CMIP5 forcings with the greatest retreat occurring in response to the GISS-E2-R

forcing. Overall the grounding line positions under the low-end parameter configuration is similar to the optimum. Mass loss and grounding line retreat is limited under this configuration due to the increased stiffness and greater basal traction, limiting delivery of ice to the grounding line and subsequent mass loss.

In comparison to the low-end and optimum parameter sets, the median and high-end parameter sets produce considerable

grounding line retreat in response to each of the CMIP5 projected sub-ice shelf melt forcings. Both parameter sets have a similarly low scaling of the ice viscosity and a high initial basal melt rate in comparison to the optimum, which is likely responsible for the greater mass loss (Nias et al., 2016). The median set of parameters results in a greater grounding line retreat over the 21st century than the high-end parameter set, despite the lower overall mass loss. This occurs because the high-end parameter set has a lower scaling factor applied to the basal traction coefficient field than the median set, producing

a more slippery bed in the former than the latter, causing increased delivery of mass toward the grounding line and offsetting grounding line retreat. Combined with softer ice and increased velocity, the relatively slippery bed also results in increased delivery of mass across the grounding line, explaining the high projected mass loss and SLE contribution of between 5.4 - 12.1 cm by 2100, despite the more muted grounding line retreat.

The response of the individual ice streams to additional melt forcing is similar for the median and high-end parameter sets. The PIG grounding line retreat is predominantly confined to its narrow embayment with considerable upstream retreat into

the main trunk. For both parameter sets, the PIG grounding line is sensitive to the magnitude of the CMIP5 ocean temperature forcing, with large differences between the final positions in 2100 across the subset. The Thwaites Glacier grounding line experiences a considerable lengthening across the wide glacier trunk for each of the CMIP5 forced experiments, in addition to an upstream retreat where the widening of the embayment has a greater control on the mass flux from the ice stream. For all parameter sets, the PSK ice streams exhibit notable grounding line retreat, controlled largely by the varying magnitudes of applied ocean forcing.

There is a significant correlation between the rate of SLE contribution and the applied CMIP5 ocean anomaly, with an $R^2$ value of >0.9 which is consistent for each of the parameter sets (fig. 6b). Whilst the response of the ASE ice streams to ocean temperature forcing is linear for each parameter set, the sensitivity to forcing is dependent on the parameter set chosen in the ice sheet model configuration, modifying both the gradient and intercept of the SLE response to temperature forcing. Moreover, the uncertainty associated with the projected SLE contribution for each AOGCM is dependent on the parameter set (fig. 6b), where models with the greatest ocean temperature forcing result in the largest range in SLE contribution when accounting for the parameter uncertainty.

## 5. Discussion

For the optimum set of parameters obtained in the initialisation procedure, we project a 2.0 - 4.5 cm SLE contribution in response to CMIP5 RCP8.5 projections of ocean temperature on-shelf in the ASE. The greater the magnitude of the temperature anomaly over the 21$^{st}$ century, the more extensive the grounding line retreat and projected mass loss from the ASE, which is consistent with findings from modelling studies and observations (Favier et al., 2014). Recent literature has established a close coupling between the basal melting of ice shelves and exacerbated grounding line retreat (Arthern and Williams, 2017; Christianson et al., 2016; Gladstone et al., 2012; Pritchard et al., 2012; Ritz et al., 2015; Seroussi et al., 2014). Given that our applied sub-shelf melt rates are derived from CMIP5 modelled ocean temperature projections, it is evident that models displaying the greatest magnitude of local warming in the ASE produce the greatest grounding line retreat and SLE by the end of the 21$^{st}$ century (Jacobs et al., 2012; Turner et al., 2017; Wåhlin et al., 2013), where large warming is likely associated with an increased volume of CDW on-shelf (Thoma et al., 2008). The varying responses to the different AOGCM forcings illustrate the dependence of the region on the sub-ice shelf melt forcing, highlighting the uncertainty in SLE projections resulting from choice of AOGCM alone.

Existing modelling investigations exploring future ASE mass evolution indicate a range of SLE contributions by the end of the 21$^{st}$ century, due to the differences in model physics and experimental design. Cornford et al., (2015) found a 1.5 to 4.0 cm SLE in response to the A1B scenario from CMIP3, which is consistent with our findings, despite the A1B scenario being of a lower magnitude forcing than RCP8.5. Furthermore, a 16 member ice sheet model intercomparison study projecting the

response to an RCP8.5 scenario by Levermann et al. (2019) gave a 90% likelihood upper bound SLE contribution of
approximately 9 cm relative to the year 2000, with a median of 2 cm. Whilst the uncertainty range  in their investigation is
derived from the differences between the ice sheet models, and thus their resolutions and model physics, the study does not
account for uncertainty associated with individual model configuration which would result in a greater uncertainty range in
SLE projections. Our projected 21$^{st}$ century sea level rise estimates are broadly consistent with existing investigations despite
the use of alternative forcing scenarios and models.

The relationship between the applied sub-ice shelf melt forcing and the rate of SLE response suggests that the ASE is
responding linearly to ocean temperature (Fig. 6b); this is consistent across the low-end, optimum, median and high-end
parameter sets. The linearity of our results would indicate that MISI is not observed in the ASE during the 21$^{st}$ century
simulations, where runaway mass loss and grounding line retreat in the region would exhibit a more nonlinear SLE
contribution.  Previous modelling studies have, however, shown that a MISI response may occur this century under very
high melt rate forcing (Arthern and Williams, 2017), or in the 22$^{nd}$ century following a perturbation applied during the 21$^{st}$
century (e.g. Martin et al., 2019). Therefore, our results do not preclude that multi-centennial MISI may have been initiated
in the simulations performed in this investigation.

We find the uncertainty associated with the ice sheet model parameters, $C$, $\varphi$ and $M_b$, obtained in the initialisation procedure
alters the sensitivity of the ASE response to ocean forced basal melting. The sensitivity of projections to uncertainties
associated with model parameters increases with increasing magnitude of ocean forcing, consistent with Bulthuis et al.
(2019). Generally, increased (decreased) viscosity, basal traction and decreased (increased) initial basal melt act to suppress
(amplify) the mass loss from the ASE ice streams and projected SLE estimates, which is illustrated by the results of the full
N16 ensemble. However, the response of the region to the perturbed basal traction parameters is not consistent with the
expected trend that has been illustrated through linear regression (Nias et al., 2016), instead perturbed parameters increase in
the order of optimum, high-end, low-end, median while the mass loss increases from low to high. This relationship may arise
partly because our experiments explore only a sample of the theoretical parameter space, whereas other, unmodelled,
parameter combinations might show clearer dependencies. However, the lack of linearity between basal traction and mass
loss may also indicate that the latter is more strongly influenced by variations in, for example, ice viscosity, than by basal
friction. The range of SLE projections in response to varied ocean forcing is therefore dependent on the specific combination
of these individual spatially varying parameters, and in our experiments, the range in SLE uncertainty attributable to
parameter selection exceeds that from choice of AOGCM forcing.

A notable deficiency with using a standalone ice sheet model lies in the inability of experiments to capture the meltwater
feedback (Donat-Magnin et al., 2017). As increased temperatures result in basal melting, the input of cold fresh water alters
ocean properties and circulation, resulting in a modification of the ocean forcing of ice shelves (Hellmer et al., 2017). The

inclusion of meltwater has been modelled to result in an increased stratification of the water column and reduction in mixing, meaning the CDW routed toward the grounding line is unmodified, resulting in enhanced melting compared with uncoupled ice-ocean model experiments (Bronselaer et al., 2018; Golledge et al., 2019). Additionally, the velocity of sub-ice shelf melt plumes, controlled by ocean circulation in addition to ice shelf cavity geometry, is influential on the sub shelf melting (Dinniman et al., 2016) and will be neglected with our simplified ocean temperature forcing. Coupling of the ice sheet model to a cavity-resolving ocean model (e.g. Naughten et al., 2018) would reduce these limitations, though at present this remains computationally expensive (Cornford et al., 2015) and thus simple ocean temperature forced experiments such as ours remain a viable approach.

## 6. Conclusions

In this investigation we use 21$^{st}$ century CMIP5 RCP8.5 projections of ocean temperature from a historically-validated subset of AOGCMs to parameterise a sub-ice shelf melt rate forcing for ice streams in the ASE. Using a set of optimum spatially varying parameters obtained from the model configuration procedure, we find a contribution to sea level rise of 2.0 - 4.5 cm by 2100, where the SLE response of the ASE is largely dependent on the choice of AOGCM forcing applied. Additional experiments using perturbed spatially varying parameter fields of basal traction, ice stiffness and initial sub shelf melt rate reveal a 12.1 cm upper bound SLE contribution for a crude 90% uncertainty associated with the configuration procedure. We find the response of the region, as shown by the projected mass loss, to be dependent largely on the magnitude of applied forcing which has been derived from projections of ocean temperature in the region. We take forward from this investigation that the perturbation of ice sheet model parameter fields has a considerable control on the projected response of the region to ocean forced basal melting, highlighting the importance of reducing uncertainty associated with ice sheet model initialisation and parameter choice.

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

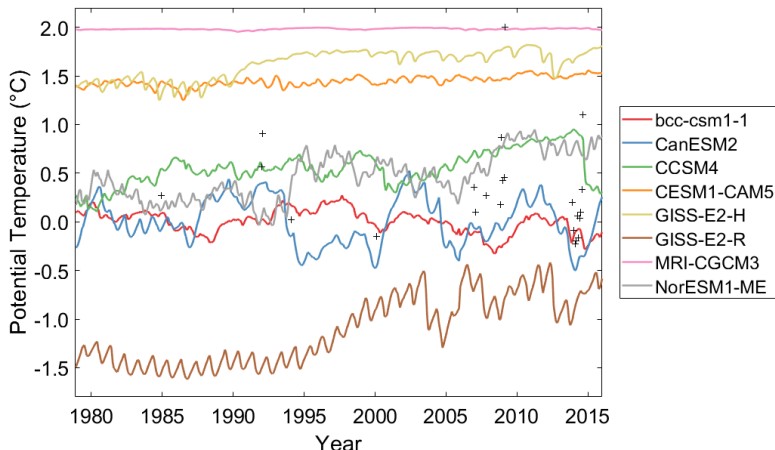

**Figure 1. Monthly mean ocean potential temperature in the ASE averaged over 400-700 m depth range produced by a subset of CMIP5 AOGCMs over the period from 1979 – 2016, where the period from 2006-2016 is made up of projections forced with RCP8.5. Black + show observed ocean potential temperature in the ASE from the Hadley Centre dataset averaged over 400-700 m depths during the same period.**

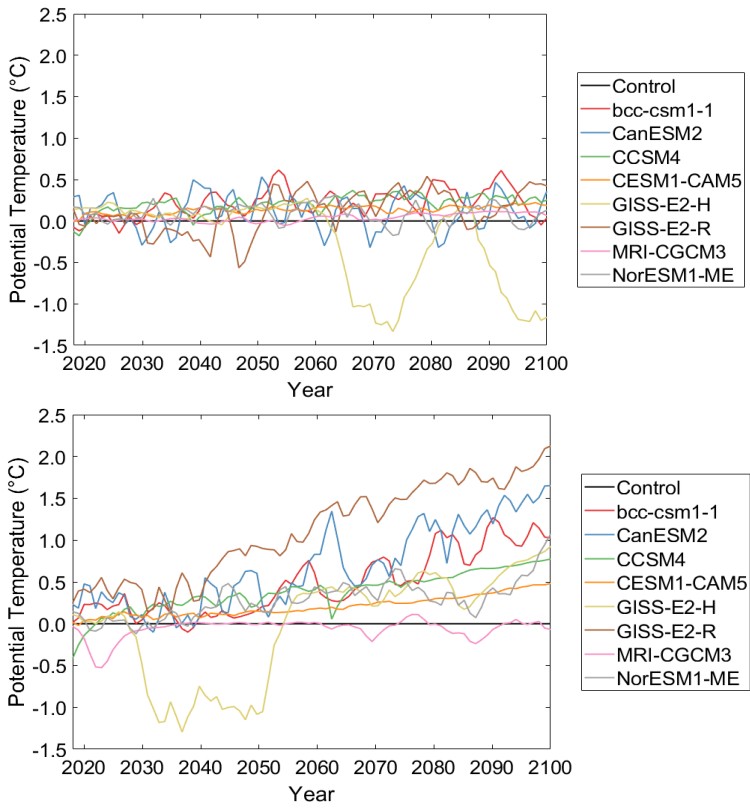

**Figure 2. Projected 21st century ASE ocean potential temperature anomalies averaged over 400-700 m depth range. Anomalies are relative to the depth averaged 400-700 m mean from 2005-2016. Each line represents a member of the CMIP5 AOGCM subset forced with the RCP8.5 (a) and RCP2.6 (b) scenarios.**

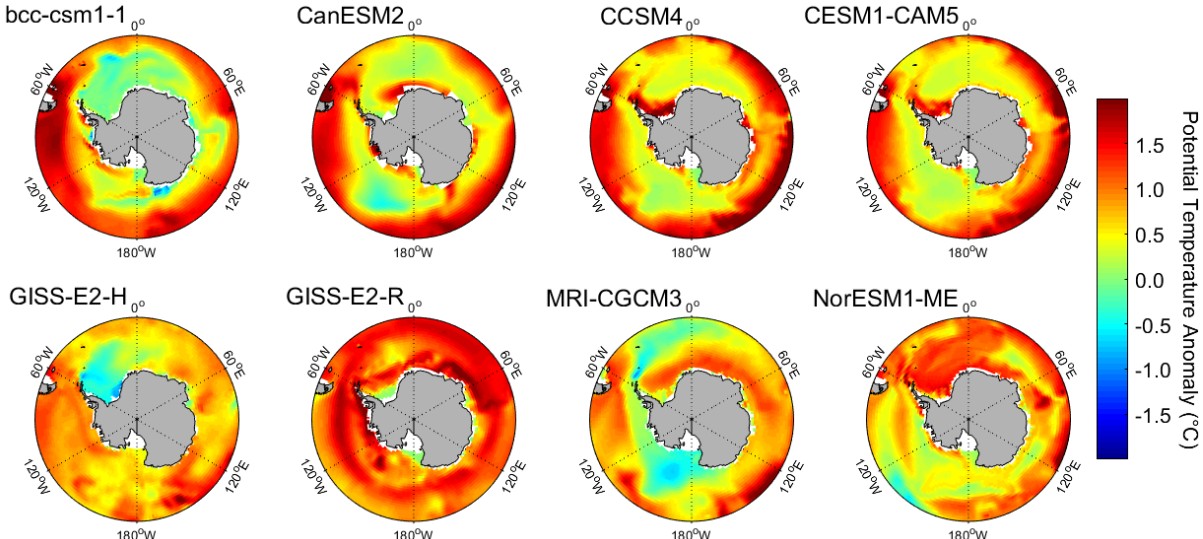

**Figure 3. Projected Southern Ocean temperature anomalies in 2100 (2091-2100 mean) averaged over 400-700 m depth range under RCP8.5 relative to the 2006-2016 mean for each of the CMIP5 AOGCM subset members.**

**Table 1. Scaling factors applied to each of the spatially varying parameter fields for the parameter sets selected from the N16 ensemble.**

|  | Basal Traction Coefficient ($C$) | Stiffening Factor ($\varphi$) | Initial Sub-Shelf Melt Rate ($M_b$) | Average Rate of SLR over 50 year transient experiment (mm/yr) |
|---|---|---|---|---|
| Low-end (B1052) | 0.662 | 0.742 | 0.730 | 0.002 |
| Optimum (B0000) | 0.500 | 0.500 | 0.500 | 0.269 |
| Median (B1016) | 0.856 | 0.218 | 0.867 | 0.316 |
| High-end (B1023) | 0.576 | 0.125 | 0.884 | 0.682 |

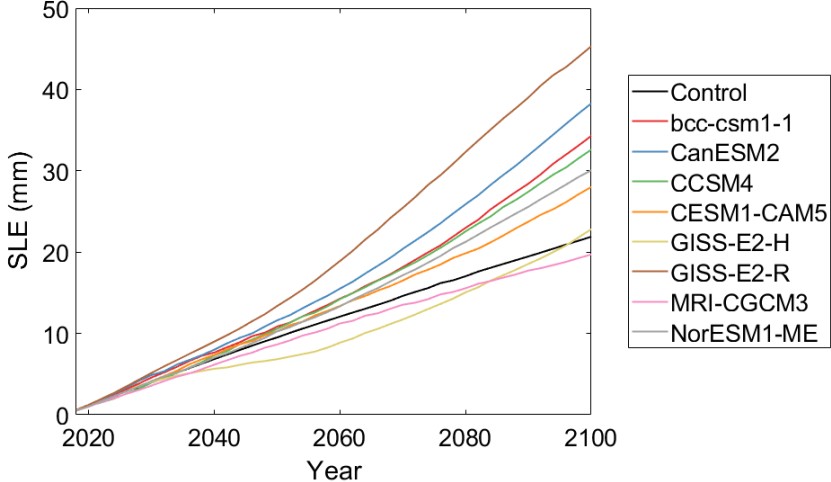

**Figure 4. Projected 21st century SLE from the ASE in response to ocean temperature forcing projected by a subset of CMIP5 AOGCMS under the RCP8.5 scenario.**

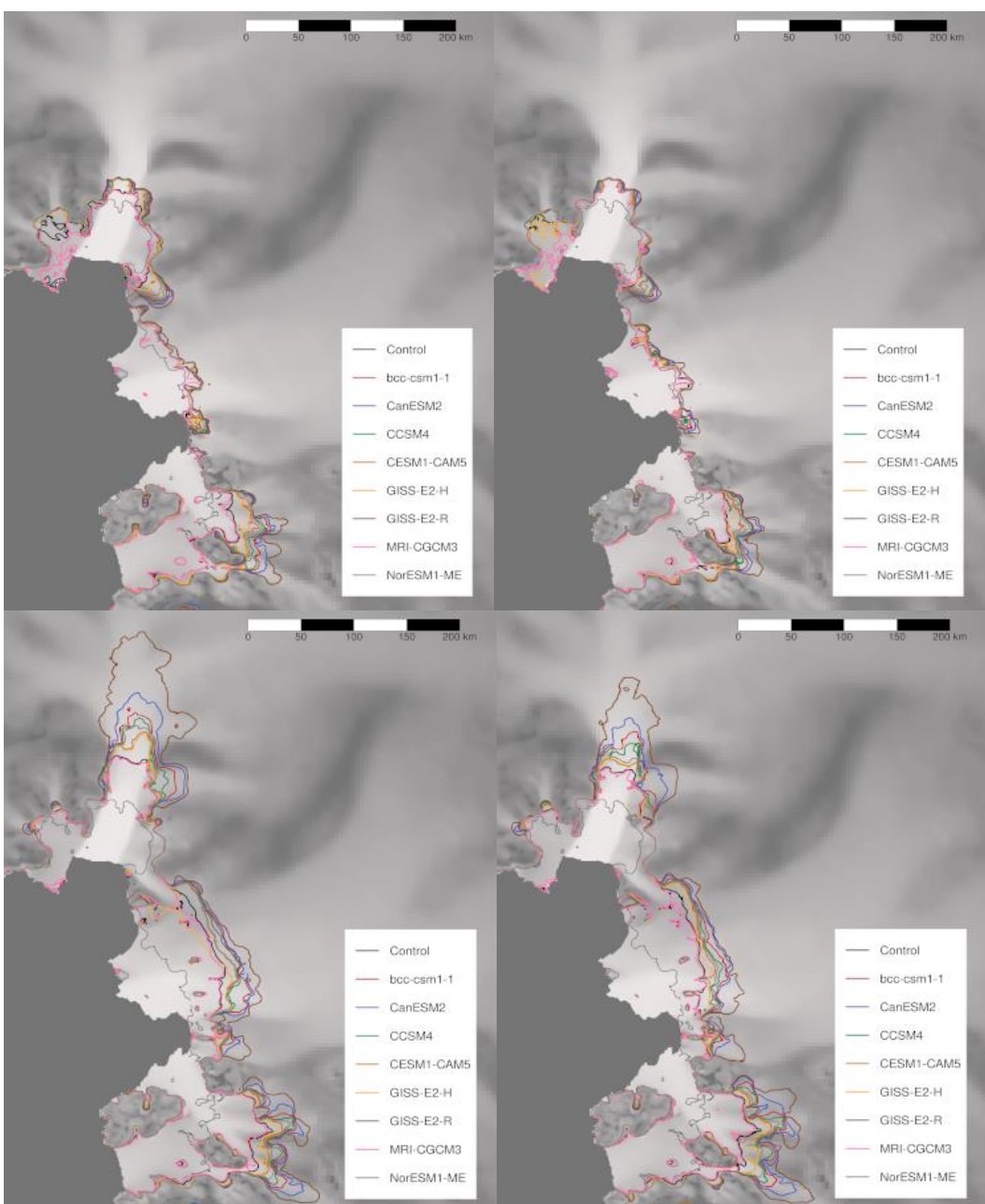

**Figure 5. ASE ice stream grounding line position in 2100 in response to each CMIP5 AOGCM projected ocean temperature forcing under RCP8.5 for each parameter set a) Optimum, b) Low-end, c) Median, d) High-end. Grey grounding line is the initial position.**

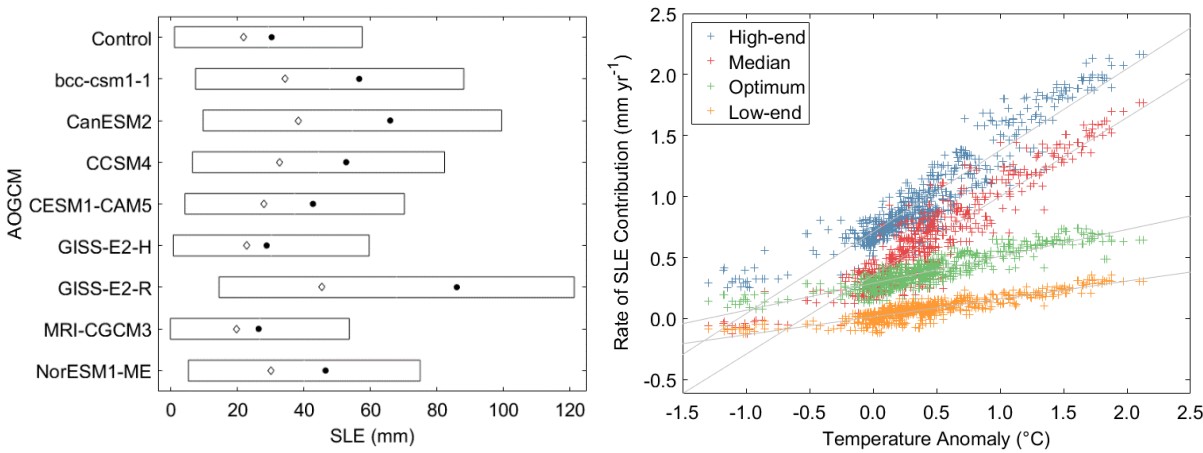

**Figure 6. a) Sea level equivalent contribution from the ASE in 2100 for each AOGCM in the subset under the RCP8.5 scenario for the range of parameter sets. Top and tail of the boxes show the high and low-end perturbed parameter sets respectively. The diamond and circle show the SLE contribution for the optimum and median parameter sets respectively. b) Rate of SLE response against ocean temperature anomaly in the ASE averaged over the 400-700 m layer over the projection period from 2017 to 2100 for each set of parameters**

**Acknowledgements**

CMIP5 output can be obtained from https://cmip.llnl.gov/cmip5/data_description.html. EN4.2.1. dataset can be obtained from https://www.metoffice.gov.uk/hadobs/en4/. BISICLES simulations were carried out on the University of Bristol's Blue Crystal Phase 3 supercomputer. BISICLES development is led by D. F. Martin at Lawrence Berkeley National Laboratory, California, USA, and S. L. Cornford at Swansea University. Data supporting the main conclusions of this study can be found at DOI 10.17605/OSF.IO/HQPS7. For the BISICLES ice sheet model spatial data in hdf5 format please contact the lead author.