# Peer review of "Ocean forced evolution of the Amundsen Sea catchment, West Antarctica, by 2100"

_The Cryosphere, 2019_

## Referee Comment (RC1) · Anonymous Referee #1 · 26 Nov 2019

The paper is a very well written and mostly well argued piece of work examining the likely response in ice VAF of the ASE area to future ocean warming, linked to RCP emission scenarios.

Whilst I do recommend publication with minor corrections, I would like to see a slightly expanded section (3.3) on how the ice shelf meltrates are derived and implemented in the ice model, in particular their choice of parameterization method.

Section 3.3

To clarify line 290.....would that be an extra 1m/a of melt for each 0.1 deg of temperature rise? And also that it is assumed that a warming in the ocean outside an (unmodelled) cavity will reach the grounding line unchanged?

[Figure]

The authors use a single, averaged mean temperature value for each model simulation to force melting on all ice shelves within the ASE. Is this a valid assumption? What is the spatial variability in oceanic conditions of the models like? Would we expect to see water with different properties entering different ice shelf cavities? Some further discussion of these points would be appreciated.

The length scale, lambda, used in the melt rate forcing has a length scale of 1000m. Is there any physical based justification of this? Is there also any reason to assume that this will be constant for all ice shelves within the domain?

The authors have used a relatively simple parameterization of melt rate that varies only with distance from the grounding line. Is there a reason they haven't used a more advanced method that would take into account changes in ice shelf basal slope? (for example a plume model such as Lazeroms et al 2018, or a box model like Reese et al 2018)

Smaller points and technical corrections

102-Is it a valid assumption that models are temporally consistent?

146 - Would it not make more sense to select models for the subset based upon their performance in the ASE rather than the Southern Ocean as a whole, given that this work is focused on the ASE.

212 How does the model deal with an advancing/retreating calving front? I assume it is held constant at the initilised position?

276 results of the

400 Pope,Smith and Kohler

500 Perhaps the discussion could also include mention of more advanced parameterization schemes?

Fig 2. The potential temperature axis seem to be labeled wrongly

**TCD**

References

Lazeroms, W. M. J., Jenkins, A., Gudmundsson, G. H., and van de Wal, R. S. W.: Modelling present-day basal melt rates for Antarctic ice shelves using a parametrization of buoyant meltwater plumes, The Cryosphere, 12, 49–70, https://doi.org/10.5194/tc-12-49-2018, 2018.

Reese, R., Albrecht, T., Mengel, M., Asay-Davis, X., and Winkelmann, R.: Antarctic sub-shelf melt rates via PICO, The Cryosphere, 12, 1969–1985, https://doi.org/10.5194/tc-12-1969-2018, 2018.

---

## Referee Comment (RC2) · Daniel Martin (Referee) · 7 Jan 2020

**1   Overview**

In this work, the authors present projections of evolution of the Amundsen Sea Catchment of the Antarctic Ice Sheet (and resulting contributions to global sea levels) under a range of climate forcings provided by CMIP5 global circulation model simulations. The specific model results used are based on assessments of model skill in the Southern Ocean, and subshelf melt-rate forcing is computed based on model-computed ocean temperature anomalies. There is also a fairly careful study of the effects of uncertainties in the initial problem setup due to the parameter estimation (via inversion) required for the initial model state to match present-day observations.

[Figure]

This is a very nice paper, for the most part well-written and clear. With some exceptions (outlined below), the methods used are clearly documented, the results are presented clearly, and the discussion is thoughtful, careful and clear. I have only minor suggestions as listed below, and would support publication once these are addressed.

**2 General Comments**

My main high-level suggestion is to consider including the dependence on initialization that you demonstrate as a key point. As a modeler, I think this is a major takeaway from this article – you demonstrate how projections can be very sensitive to something which is almost always only mentioned in passing and can be something of a dark art. You even make it the final sentence of the paper, which indicates its significance.

After reading the article a few times and also looking at Nias et al (2016), I admit that I'm confused as to whether you're running full-continent AIS simulations while forcing only in the ASE sector, or if you're running a regional model. I think it's the former, but that should be clarified.

Graphs in my preprint are too small to read easily when printed on paper. They should be larger in a final printed version.

**3 Specific Comments**

1. line 55: Seroussi and Morlighem (2018) is an odd reference to use here, since that work focused primarily on evaluating approaches to discretizing subshelf melt in ice sheet models, not the effects of subshelf melting in general.

2. line 219: I'd suggest the use of "finest" instead of "maximum" when discussing

resolution.

3. line 253: I don't think "accounted for" is the right way to say this. Perhaps "addressed by"?

4. line 257: What is $r$? Did you mean $b$ (bedrock)? (would it be simpler to simply say "grounded" and "floating"?)

5. section 3.2: As I mentioned in the general comments above, I *think* you're running on a whole-continent AIS domain, but I don't think you explicitly say that anywhere (apologies if I missed it). In this section, it would be helpful if you can provide the following:

    (a) Explicitly state which domain you're solving on (a picture with your initial velocity field might be helpful here).
    (b) Are you modifying the state outside the ASE in any way? (For example, in other works, BISICLES has been run with high friction values imposed outside the catchment of interest to "turn off" the flow there, isolating the flow to the single catchment – are you doing anything like that here?)
    (c) What are you doing for marine forcing outside the ASE?

6. line 356: You might also consider citing the recent paper by Gudmundsson, et al (2019) here.

7. line 366: I'd suggest rephrasing "...grounding line retreat in 2100 relative to...", possibly as "grounding line positions in 2100 which are retreated relative to..."

8. line 367: I'd suggest rephrasing "The individual ice stream response...", possibly as "Response of the individual ice streams..."

9. lines 489-513: This paragraph doesn't read as well as the rest of the paper and could use some editing for clarity.

10. line 492: This sentence is a bit convoluted. I think you meant to say "instead performing" instead of "instead of performing", perhaps?

11. line 495: In fairness, our catchment-independence results in the 2019 GRL paper suggest that you're safe doing basin-scale models for O(100 years) or so.

12. line 526: As I mentioned above, I think this is a key point of your work here (and should be consequently included in the "key points" at the beginning).

13. Table 1: It's odd that the basal traction isn't monotone as you progress from low-end through optimum, median, and high-end, the way all of the other parameters are. It might be helpful to comment a bit more on that. Are there other ensemble members which have similar "high-end" responses while also being monotone in all of the quantities, relative to the other cases? As you discuss in the text, this case is interesting because it helps set off the competing impacts of varying friction vs. viscosity, but on the other hand, it might obscure more basic relationships in these quantities.

14. Figure 5: It would be useful if you labeled various ice streams (and other features) referred to in the text in at least one frame of this figure.

**4 Technical Comments**

1. line 50: Formatting is odd here, with 2 words and the remainder empty space. Perhaps an errant line break?

2. line 199: missing period at the end of the sentence.

3. line 224: The formatting of this equation is off – the dot for the divergence appears to be a period. If you're using latex, use the `\cdot` character; in MS Word, there is a dot-product dot in the math character set.

4. line 241 (eqn 3): More equation formatting. Along with the dot-product issue already mentioned above, I'm confused by the character above the viscosity ($\mu$). I think it's meant to be the overbar (indicating vertical averaging?). Finally, there is an accent over $\epsilon$, rather than what's generally a time-derivative dot. (in latex, `\dot{\epsilon}` produces $\dot{\epsilon}$. In MS-Word, there is a way to do that using equation formatting.)

5. line 248: There should be a comma between "n=3" and "satisfies".

6. line 257: Formatting for this set of cases is a bit off (too crowded, vertically) – this is true for equations 6,7, and 8.

7. line 269: Same issue with the dot-product formatting.

8. line 272: "...above, BISICLES..." → "...above, the BISICLES..."

9. line 276: "...results of initialization" → "...results of an initialization..."

10. line 284: I'd suggest "consistent with present day observations"

11. line 301: I don't think you need (or want) a comma after "forcing"

12. line 368: "response... differs"

13. line 378: I don't think "denote" is the right word here; perhaps "indicate"?

14. line 393: should it be "has a lesser", or "have a lesser"?

15. line 400: I'd suggest commas after "Pope" and "Smith"

16. line 450: I'd suggest "response" instead of "behaviour"

17. line 463: Do you mean fig 6b here?

18. line 479-484: This is a pretty long sentence...

19. line 482: "models... produce"

20. line 483: I think the semicolon should be a comma here, since what comes after isn't an independent clause...

21. line 485: "responses... illustrate"

22. line 486: (same thing) "responses ... highlight"

23. lines 498-499: I think this is a sentence fragment.

24. line 508: "captured by the within model configuration"?

25. line 512: "our range... is marginally..."

26. line 516: Do you mean fig 6b here?

27. line 550: "cavity resolving" → "cavity-resolving", perhaps?

28. Figure 6 caption: Did you mean the "400-00m layer" as stated, or is there a typo?

**5  References**

1. G. Hilmar Gudmundsson, Fernando S. Paolo, Susheel Adusumilli, and He- len A. Fricker. "Instantaneous Antarctic ice sheet mass loss driven by thinning ice shelves". *Geophysical Research Letters*, **46**(23):13903–13909, 2019.

---

## Author Comment (AC1) · 14 Jan 2020

Section 3.3 To clarify line 290.....would that be an extra 1m/a of melt for each 0.1 deg of temperature rise? And also that it is assumed that a warming in the ocean outside an (unmodelled) cavity will reach the grounding line unchanged? The authors use a single, averaged mean temperature value for each model simulation to force melting on all ice shelves within the ASE. Is this a valid assumption? What is the spatial variability in oceanic conditions of the models like? Would we expect to see water with different properties entering different ice shelf cavities? Some further discussion of these points would be appreciated. »» We have added the following sentences to section 3.3 to address these questions: "...where an additional 0.1°C temperature increase results in an increase of 1 m/a to the basal melt rate. The CMIP5 AOGCM forcing data that we

use are relatively coarse in their spatial resolution and also do not capture sub-ice shelf oceanographic conditions. Consequently, we are unable to accurately incorporate the spatial complexity of ocean temperature variability that exists in the ASE (c.f. Turner et al., 2017). Given that our input data better reflect regional rather than local-scale oceanic changes, we force our simulations with spatially-averaged CMIP5 temperature anomalies."

The length scale, lambda, used in the melt rate forcing has a length scale of 1000m. Is there any physical based justification of this? Is there also any reason to assume that this will be constant for all ice shelves within the domain? »» There was a mistake in the text and the length scale is 10,000m which has now been amended (line 305). This length scale follows the practice of Cornford et al. (2015) and we are not aware of the physical basis for this.

The authors have used a relatively simple parameterization of melt rate that varies only with distance from the grounding line. Is there a reason they haven't used a more advanced method that would take into account changes in ice shelf basal slope? (for example a plume model such as Lazeroms et al 2018, or a box model like Reese et al 2018) »» To address this we have added the following paragraph to section 3.3: "The simplified distance dependent melt parameterisation employed in this investigation was chosen in order to maintain continuity with the Nias et al., (2016; 2019) studies. Our parameterisation neglects the effect of overturning circulation within an ice shelf cavity in addition to the ice shelf cavity geometry and presence of meltwater plumes which influence the pattern of sub-ice shelf basal melting (Dinniman et al., 2016). Whilst more complex parameterisations attempt to incorporate these mechanisms (e.g. Lazeroms et al., 2018; Reese et al., 2018), no parameterisation is yet able to replicate known patterns of sub-ice shelf melting (Favier et al., 2019). Furthermore, the uncertainty associated with the magnitude of the future forcing exceeds that associated with the parameterisation of sub-shelf melting (Holland et al., 2019), justifying the use of the simplified parameterisation employed in this investigation."

Smaller points and technical corrections 102-Is it a valid assumption that models are temporally consistent? »» Yes, it is a valid assumption that model performance is temporally consistent.

146 - Would it not make more sense to select models for the subset based upon their performance in the ASE rather than the Southern Ocean as a whole, given that this work is focused on the ASE. »» Already explained in lines 117-120.

212 How does the model deal with an advancing/retreating calving front? I assume it is held constant at the initilised position? »» Yes- fixed calving front. This has been added to 286.

276 results of the »» Done- changed to 'an' as suggested by Reviewer 2

400 Pope,Smith and Kohler »» Done (401)

500 Perhaps the discussion could also include mention of more advanced parameterization schemes? »» See lines 550-556.

Fig 2. The potential temperature axis seem to be labeled wrongly »» Thank you for identifying this, it has been corrected.
* * *
[Figure]

**Fig. 1.** Updated figure 2a (see revised manuscript for full caption).

---

## Author Comment (AC2) · 14 Jan 2020

In this work, the authors present projections of evolution of the Amundsen Sea Catchment of the Antarctic Ice Sheet (and resulting contributions to global sea levels) under a range of climate forcings provided by CMIP5 global circulation model simulations. The specific model results used are based on assessments of model skill in the Southern Ocean, and subshelf melt-rate forcing is computed based on model-computed ocean temperature anomalies. There is also a fairly careful study of the effects of uncertainties in the initial problem setup due to the parameter estimation (via inversion) required for the initial model state to match present-day observations.

This is a very nice paper, for the most part well-written and clear. With some exceptions

(outlined below), the methods used are clearly documented, the results are presented clearly, and the discussion is thoughtful, careful and clear. I have only minor suggestions as listed below, and would support publication once these are addressed.

2 General Comments

My main high-level suggestion is to consider including the dependence on initialization that you demonstrate as a key point. As a modeler, I think this is a major takeaway from this article – you demonstrate how projections can be very sensitive to something which is almost always only mentioned in passing and can be something of a dark art. You even make it the final sentence of the paper, which indicates its significance. After reading the article a few times and also looking at Nias et al (2016), I admit that I'm confused as to whether you're running full-continent AIS simulations while forcing only in the ASE sector, or if you're running a regional model. I think it's the former, but that should be clarified.

Graphs in my preprint are too small to read easily when printed on paper. They should be larger in a final printed version. »» The font sizes of the figures have been increased and are included in the revised manuscript.

Specific Comments 1. line 55: Seroussi and Morlighem (2018) is an odd reference to use here, since that work focused primarily on evaluating approaches to discretizing subshelf melt in ice sheet models, not the effects of subshelf melting in general. »» Thank you for pointing this out. The reference has been amended to Schoof (2007).

2. line 219: I'd suggest the use of "finest" instead of "maximum" when discussing resolution. »» Done.

3. line 253: I don't think "accounted for" is the right way to say this. Perhaps "addressed by"? »» Done.

4. line 257: What is r? Did you mean b (bedrock)? (would it be simpler to simply say "grounded" and "floating"?) »» Thank you for noticing, this should be bedrock b and

has been changed.

5. section 3.2: As I mentioned in the general comments above, I think you're running on a whole-continent AIS domain, but I don't think you explicitly say that anywhere (apologies if I missed it). In this section, it would be helpful if you can provide the following: (a) Explicitly state which domain you're solving on (a picture with your initial velocity field might be helpful here). (b) Are you modifying the state outside the ASE in any way? (For example, in other works, BISICLES has been run with high friction values imposed outside the catchment of interest to "turn off" the flow there, isolating the flow to the single catchment – are you doing anything like that here?) (c) What are you doing for marine forcing outside the ASE? »» We have clarified this in line 354 of the updated manuscript: "We perform regional simulations of the ASE sector on the domain defined in Cornford et al. (2015)."

6. line 356: You might also consider citing the recent paper by Gudmundsson, et al (2019) here. »» We have included this reference, thank you for the suggestion.

7. line 366: I'd suggest rephrasing "...grounding line retreat in 2100 relative to...", possibly as "grounding line positions in 2100 which are retreated relative to..." »» Done

8. line 367: I'd suggest rephrasing "The individual ice stream response...", possibly as "Response of the individual ice streams..." »» Done

9. lines 489-513: This paragraph doesn't read as well as the rest of the paper and could use some editing for clarity. »» This paragraph has been edited: "Existing modelling investigations exploring future ASE mass evolution indicate a range of SLE contributions by the end of the 21st century, due to the differences in model physics and experimental design. Cornford et al., (2015) found a 1.5 to 4.0 cm SLE in response to the A1B scenario from CMIP3, which is consistent with our findings, despite the A1B scenario being of a lower magnitude forcing than RCP8.5. Furthermore, a 16 member ice sheet model intercomparison study projecting the response to an RCP8.5 scenario by Levermann et al. (2019) gave a 90% likelihood upper bound SLE contribution of approximately 9

cm relative to the year 2000, with a median of 2 cm. Whilst the uncertainty range in their investigation is derived from the differences between the ice sheet models, and thus their resolutions and model physics, the study does not account for uncertainty associated with individual model configuration which would result in a greater uncertainty range in SLE projections. Our projected 21st century sea level rise estimates are broadly consistent with existing investigations despite the use of alternative forcing scenarios and models."

10. line 492: This sentence is a bit convoluted. I think you meant to say "instead performing" instead of "instead of performing", perhaps? »» This sentence has been removed in the editing of the paragraph (as proposed in comment 9).

11. line 495: In fairness, our catchment-independence results in the 2019 GRL paper suggest that you're safe doing basin-scale models for O(100 years) or so. »» This sentence has been removed in the editing of the paragraph (as proposed in comment 9).

12. line 526: As I mentioned above, I think this is a key point of your work here (and should be consequently included in the "key points" at the beginning). »» We accept this suggestion and the sentence has been added to the key points.

13. Table 1: It's odd that the basal traction isn't monotone as you progress from low end through optimum, median, and high-end, the way all of the other parameters are. It might be helpful to comment a bit more on that. Are there other ensemble members which have similar "high-end" responses while also being monotone in all of the quantities, relative to the other cases? As you discuss in the text, this case is interesting because it helps set off the competing impacts of varying friction vs. viscosity, but on the other hand, it might obscure more basic relationships in these quantities. »» We have amended the following discussion paragraph to incorporate this suggestion and hope it sufficiently addresses this comment. "We find the uncertainty associated with the ice sheet model parameters, C, $\varphi$ and M_b, obtained in the initialisation procedure

alters the sensitivity of the ASE response to ocean forced basal melting. The sensitivity of projections to uncertainties associated with model parameters increases with increasing magnitude of ocean forcing, consistent with Bulthuis et al. (2019). Generally, increased (decreased) viscosity, basal traction and decreased (increased) initial basal melt act to suppress (amplify) the mass loss from the ASE ice streams and projected SLE estimates, which is illustrated by the results of the full N16 ensemble. However, the response of the region to the perturbed basal traction parameters is not consistent with the expected trend that has been illustrated through linear regression (Nias et al., 2016), instead perturbed parameters increase in the order of optimum, high-end, low-end, median while the mass loss increases from low to high. This relationship may arise partly because our experiments explore only a sample of the theoretical parameter space, whereas other, unmodelled, parameter combinations might show clearer dependencies. However, the lack of linearity between basal traction and mass loss may also indicate that the latter is more strongly influenced by variations in, for example, ice viscosity, than by basal friction. The range of SLE projections in response to varied ocean forcing is therefore dependent on the specific combination of these individual spatially varying parameters, and in our experiments, the range in SLE uncertainty attributable to parameter selection exceeds that from choice of AOGCM forcing."

14. Figure 5: It would be useful if you labeled various ice streams (and other features) referred to in the text in at least one frame of this figure. »» Labels have been added.

Technical Comments 1. line 50: Formatting is odd here, with 2 words and the remainder empty space. Perhaps an errant line break? »» Amended.

2. line 199: missing period at the end of the sentence. »» Now added.

3. line 224: The formatting of this equation is off – the dot for the divergence appears to be a period. If you're using latex, use the \cdot character; in MS Word, there is a dot-product dot in the math character set. »» Amended.

4. line 241 (eqn 3): More equation formatting. Along with the dot-product issue already

mentioned above, I'm confused by the character above the viscosity ($\mu$). I think it's meant to be the overbar (indicating vertical averaging?). Finally, there is an accent over , rather than what's generally a time-derivative dot. (in latex, \dot{\epsilon} produces ËŹ. In MS-Word, there is a way to do that using equation formatting.) »» This has been changed now, thank you.

5. line 248: There should be a comma between "n=3" and "satisfies". »» Done

6. line 257: Formatting for this set of cases is a bit off (too crowded, vertically) – this is true for equations 6,7, and 8. »» These equations have now been modified.

7. line 269: Same issue with the dot-product formatting. »» Done.

8. line 272: "...above, BISICLES..." → "...above, the BISICLES..." »» Done.

9. line 276: "...results of initialization" → "...results of an initialization..." »» Done.

10. line 284: I'd suggest "consistent with present day observations" »» Done.

11. line 301: I don't think you need (or want) a comma after "forcing" »» Removed.

12. line 368: "response... differs" »» Changed

13. line 378: I don't think "denote" is the right word here; perhaps "indicate"? »» Changed

14. line 393: should it be "has a lesser", or "have a lesser"? »» Changed to have

15. line 400: I'd suggest commas after "Pope" and "Smith" »» Commas added

16. line 450: I'd suggest "response" instead of "behaviour" »» Changed

17. line 463: Do you mean fig 6b here? »» Yes, this has been changed

18. line 479-484: This is a pretty long sentence... »» Shortening of this sentence has been considered but the authors have decided to keep it unchanged.

19. line 482: "models... produce" »» Changed

20. line 483: I think the semicolon should be a comma here, since what comes after isn't an independent clause... »» Changed to a comma.

21. line 485: "responses... illustrate" »» Changed

22. line 486: (same thing) "responses ... highlight" »» Changed to '...melt forcing, highlighting'

23. lines 498-499: I think this is a sentence fragment. ?? »» This has been considered and the authors have decided to keep this sentence unchanged.

24. line 508: "captured by the within model configuration"? »» This sentence has been changed in the editing of the paragraph (as proposed in comment 9).

25. line 512: "our range... is marginally..." »» Changed.

26. line 516: Do you mean fig 6b here? »» Changed

27. line 550: "cavity resolving" → "cavity-resolving", perhaps? »» Changed.

28. Figure 6 caption: Did you mean the "400-00m layer" as stated, or is there a typo? »» This is a typo, thank you for identifying it. It has been amended.

---

## Author Response (AR1)

[revised manuscript text omitted]
. Often, studies tend to split continental scale simulations into catchments, instead of performing catchment scale simulations and therefore boundaries of the ASE region tend to vary, making comparison of SLE contribution projections challenging. Furthermore, catchment scale simulations of this kind will neglect the interactions between catchments that will be present in continental scale simulations~~
415 ~~(Martin et al., 2019). Cornford et al., (2015) found a 1.5 to 4.0 cm SLE in response to the A1B scenario from CMIP3, which is consistent with our findings, despite the A1B scenario being of a lower magnitude forcing than RCP8.5. In contrast, an ASE upper bound of 25 cm SLE by 2100 (95% quantile) estimates for A1B scenario forcing (Ritz et al., 2015), which is over double our projected upper bound. The same study presented a 50% likelihood probability of the ASE contribution not exceeding 7.5 cm and a modal projection of 2.2 cm (Ritz et al., 2015). Whilst the upper limit of sea level rise well exceeds~~
420 ~~the equivalent value from our results, the more probabilistically likely values from their investigation are closer to the projections we present. Meanwhile, a 16 member ice sheet model intercomparison study projecting the response to an RCP8.5 scenario by Levermann et al. (2019) gave a 90% likelihood upper bound SLE contribution of approximately 9 cm relative to the year 2000, with a median of 2 cm. Whilst the range in uncertainty in their investigation is derived from the differences between the ice sheet models, and thus their resolutions and model physics, there is no uncertainty captured by~~
425 ~~the within model configuration which could result in a greater uncertainty range in SLE projections. Although there appears to be some consistency with the projection of SLE contribution by 2100 established in the aforementioned investigations, by capturing some of the uncertainty associated with the ocean forcing, our range in estimates of 2.0 – 4.5 cm are marginally greater than those projected in previous studies.~~

430 The relationship between the applied sub-ice shelf melt forcing and the rate of SLE response suggests that the ASE is responding linearly to ocean temperature (Fig. 6b5a); this is consistent across the low-end, optimum, median and high-end parameter sets. The linearity of our results would indicate that MISI is not observed in the ASE during the 21ˢᵗ century simulations, where runaway mass loss and grounding line retreat in the region would exhibit a more nonlinear SLE contribution. Previous modelling studies have, however, shown that a MISI response may occur this century under very
435 high melt rate forcing (Arthern and Williams, 2017), or in the 22ⁿᵈ century following a perturbation applied during the 21ˢᵗ century (e.g. Martin et al., 2019). Therefore, our results do not preclude that multi-centennial MISI may have been initiated in the simulations performed in this investigation.

We find the uncertainty associated with the ice sheet model parameters, $C$, $\varphi$ and $M_b$, obtained in the initialisation procedure
440 alters the sensitivity of the ASE response to ocean forced basal melting. The sensitivity of projections to uncertainties associated with model parameters increases with increasing magnitude of ocean forcing, consistent with Bulthuis et al.

(2019). Generally, increased (decreased) viscosity, basal traction and decreased (increased) initial basal melt act to suppress (amplify) the mass loss from the ASE ice streams and projected SLE estimates, which is illustrated by the results of the full N16 ensemble. However, the response of the region to the perturbed basal traction parameters is not consistent with the expected trend that has been illustrated through linear regression (Nias et al., 2016), instead perturbed parameters increase in the order of optimum, high-end, low-end, median while the mass loss increases from low to high. This relationship may arise partly because our experiments explore only a sample of the theoretical parameter space, whereas other, unmodelled, parameter combinations might show clearer dependencies. However, the lack of linearity between basal traction and mass loss may also indicate that the latter is more strongly influenced by variations in, for example, ice viscosity, than by basal friction. The range of SLE projections in response to varied ocean forcing is therefore dependent on the specific combination of these individual spatially varying parameters, and in our experiments, the range in SLE uncertainty attributable to parameter selection exceeds that from choice of AOGCM forcing.

. However, the varying combinations of each perturbation means that this is not consistent across the ensemble and therefore the direct relationship between perturbations to each individual parameter and the resulting impact on grounding line migration and VAF loss cannot be discerned with this data alone. The range in 
[revised manuscript text omitted]